# In Vivo Evaluation of (−)-Zampanolide Demonstrates Potent and Persistent Antitumor Efficacy When Targeted to the Tumor Site

**DOI:** 10.3390/molecules27134244

**Published:** 2022-07-01

**Authors:** Leila Takahashi-Ruiz, Joseph D. Morris, Phillip Crews, Tyler A. Johnson, April L. Risinger

**Affiliations:** 1Department of Pharmacology, Mays Cancer Center, University of Texas Health Science Center San Antonio, San Antonio, TX 78229, USA; takahashil@livemail.uthscsa.edu; 2Department of Natural Sciences and Mathematics, Dominican University of California, San Rafael, CA 94901, USA; joseph.morris@dominican.edu; 3Department of Chemistry and Biochemistry, University of California Santa Cruz, Santa Cruz, CA 95064, USA; pcrews@ucsc.edu

**Keywords:** triple-negative breast cancer, microtubule stabilizers, zampanolide, paclitaxel, covalent

## Abstract

Microtubule-stabilizing agents (MSAs) are a class of compounds used in the treatment of triple-negative breast cancer (TNBC), a subtype of breast cancer where chemotherapy remains the standard-of-care for patients. Taxanes like paclitaxel and docetaxel have demonstrated efficacy against TNBC in the clinic, however new classes of MSAs need to be identified due to the rise of taxane resistance in patients. (−)-Zampanolide is a covalent microtubule stabilizer that can circumvent taxane resistance in vitro but has not been evaluated for in vivo antitumor efficacy. Here, we determine that (−)-zampanolide has similar potency and efficacy to paclitaxel in TNBC cell lines, but is significantly more persistent due to its covalent binding. We also provide the first reported in vivo antitumor evaluation of (−)-zampanolide where we determine that it has potent and persistent antitumor efficacy when delivered intratumorally. Future work on zampanolide to further evaluate its pharmacophore and determine ways to improve its systemic therapeutic window would make this compound a potential candidate for clinical development through its ability to circumvent taxane-resistance mechanisms.

## 1. Introduction

Breast cancer is the most common cancer diagnosed among women in the United States and is the second leading cause of cancer death [1]. Among the different types of breast cancer, triple negative breast cancer (TNBC) accounts for 15–20% of all breast cancers [2] and requires an aggressive clinical course because of the high metastatic potential with elevated rates of relapse and lower survival than other breast cancers [1]. Chemotherapy remains the standard-of-care for patients with TNBC and some of the most effective chemotherapeutics are microtubule-stabilizing agents (MSAs). MSAs decrease microtubule dynamicity, which inhibits the formation of the mitotic spindle leading to antimitotic effects and disruption of other microtubule-dependent cellular processes including protein transport, cellular signaling, and maintenance of cell shape [3]. The taxanes are a class of MSAs that include the natural product paclitaxel and the semi-synthetic derivative docetaxel. These taxanes are commonly used for the treatment of TNBC, however their efficacy is hampered by drug resistance, which often occurs through upregulation of drug efflux pumps or alterations in tubulin isotype expression [4]. The efficacy of the taxanes in TNBC demonstrates that microtubule stabilization is a validated mechanism of treating this disease. However, there is a need to identify new classes of MSAs that retain efficacy in clinically relevant models of taxane resistance.

(−)-Zampanolide (referred to as zampanolide in the current manuscript) is a 20-membered macrolide isolated from marine sponges *Fasciospongia rimosa* and *Cacospongia mycofijiensis* where the absolute stereochemistry was determined by enantioselective total synthesis to be 11*S*, 15*S*, 19*S*, 20*S* [5,6]. Zampanolide is a MSA that binds within the taxane-site on β-tubulin but is distinct from the taxanes in that it binds covalently and irreversibly to microtubules. Zampanolide promotes microtubule stabilization and G_2_/M arrest in hematological and solid tumor models in vitro, leading to antiproliferative and cytotoxic effects at low nanomolar concentrations [5]. The mechanism of microtubule stabilization of zampanolide downstream of microtubule binding is shared with paclitaxel in that they both promote the reorganization of the disordered S7-H9 M-loop into an α-helix, which leads to helix–helix interactions and enhanced stabilization of the microtubule [7]. However, the covalent binding between the C9 of zampanolide with H229 on β-tubulin makes this microtubule stabilization more persistent than taxane-mediated stabilization, which allows zampanolide to retain efficacy in taxane resistant models in vitro [5,7,8,9,10].

Covalent binding to the drug target is a validated mechanism to overcome drug resistance and this is a shared feature among other microtubule stabilizers that irreversibly bind to tubulin, including cyclostreptin and the taccalonolides. Cyclostreptin was the first MSA discovered to covalently bind to tubulin [11]. Although cyclostreptin maintains efficacy in P-glycoprotein [11] and βIII-tubulin [9] expressing cells due to its covalent binding, chemical instability and low cellular potency made it a less favorable compound for further investigation. The C22,23-epoxytaccalonolides also bind covalently to β-tubulin [12] to promote microtubule stabilization with in vitro potency similar to the taxanes, leading to antimitotic and cytotoxic effects [13,14]. While the taccalonolides retain efficacy in taxane-resistant models both in vitro and in vivo, they have a narrow therapeutic index with systemic administration [9,15]. In spite of their shared covalent binding to β-tubulin [9,16] these drugs have distinct effects on microtubule stabilization; zampanolide stabilizes the M-loop of β-tubulin in a manner similar to paclitaxel while taccalonolide and cyclostreptin binding is not associated with M-loop stabilization [9,13,14]. This is consistent with differences between the effects of these drugs on the kinetics of microtubule polymerization in biochemical assays where zampanolide and paclitaxel promote an immediate stabilization in contrast to the lag in polymerization associated with cyclostreptin or the taccalonolides [9].

These data demonstrate that zampanolide has a unique mechanism of microtubule stabilization involving M-loop stabilization through covalent binding that is distinct from both the taxanes and other covalent stabilizers. We hypothesize that zampanolide could be a prime candidate for the treatment of TNBC because it has the advantage of taxane-like M-loop stabilization, which has a known clinical benefit, through irreversible binding to retain efficacy in taxane resistant settings. However, due to the scarcity of the zampanolide natural product, the compound has not previously been evaluated for in vivo efficacy in any model. Herein we describe the isolation of zampanolide and the evaluation of its potency and efficacy in a panel of molecularly diverse TNBC cell lines in vitro, including its strong degree of cellular persistence as compared to paclitaxel. Most importantly, we demonstrate, for the first time, the potent anti-tumor efficacy of zampanolide in a TNBC xenograft model, albeit with a narrow therapeutic window that required local administration.

## 2. Results

### 2.1. Tubulin Stabilization by Zampanolide

To directly compare the microtubule-stabilizing kinetics of zampanolide to other classes of microtubule-targeted drugs, we performed a biochemical assay to monitor the polymerization of purified tubulin heterodimers into microtubules. This assay takes advantage of the fact that microtubules are cold-labile, allowing for real-time evaluation of the kinetics of microtubule polymerization when samples are warmed from 4 to 37 °C. While tubulin polymerizes on its own under these conditions, the rate and extent of this polymerization is enhanced by microtubule stabilizers, such as paclitaxel and zampanolide, and suppressed by microtubule destabilizers, such as combretastatin A-4 (Figure 1). Consistent with previous reports, zampanolide promoted the immediate polymerization of purified tubulin with similar kinetics to those induced by paclitaxel (Figure 1). In contrast, the polymerization of tubulin was delayed in the presence of taccalonolide AF, which does not promote strong M-loop stabilization [13]. These data align with the known mechanisms of action of each MSA and highlight that zampanolide and the taccalonolides promote microtubule stabilization through distinct mechanisms despite the fact that they both covalently bind to their tubulin target.

### 2.2. In Vitro Activity of Zampanolide in TNBC Cell Lines

While zampanolide has shown efficacy in drug sensitive and resistant ovarian, cervical, leukemia, and prostate cancer models, its evaluation in breast cancer lines has been confined to the ER positive MCF-7 model. Since MSAs are a mainstay in the treatment of TNBCs, the antiproliferative and cytotoxic effects of zampanolide were evaluated in a panel of six molecularly distinct TNBC cancer cell lines using the sulforhodamine B (SRB) assay. Zampanolide demonstrated low nanomolar antiproliferative potency in each of the six TNBC cell lines with concentrations that inhibited growth by 50% (GI_50_ values) ranging from 2.8–5.4 nM (Table 1). These values are comparable to the in vitro potency of paclitaxel and consistent with the single digit antiproliferative potency observed for zampanolide against other cancer cells in vitro [5,16,17,18]. However, differential cytotoxic efficacy of zampanolide was observed across TNBC lines, which is measured by comparing the final cellular density to the density at the time of drug addition (represented by the dashed line at y = 0 in Figure 2). While primarily antiproliferative effects were observed in the HCC1937, BT-549, and MDA-MB-231 TNBC lines, cytotoxicity was clearly observed in the HCC1806, HCC70, and MDA-MB-453 TNBC lines as evidenced by the curves falling below the measure of cellular density at the time of drug addition (Figure 2). This relative cytotoxic efficacy among TNBC lines is consistent with other microtubule stabilizing and destabilizing agents [19], suggesting that zampanolide does not demonstrate any distinct advantages or disadvantages in its cytotoxic efficacy against TNBC models as compared to clinically approved drugs of this class.

The mechanistic effects of zampanolide on cellular microtubules were evaluated in HCC1937 TNBC cells by indirect immunofluorescence. As expected, zampanolide promoted an increase in the density of interphase microtubules as evidenced by the reorganization of the microtubule cytoskeleton into thick microtubule bundles (Figure 3A). Additionally, cells going through mitosis also demonstrated a distinct microtubule morphology where the normal bipolar mitotic spindle is replaced with numerous microtubule asters (Figure 3B). The absence of a bipolar spindle leads to disorganization of the chromosomes, mitotic arrest, and improper segregation of chromosomes into any cells that undergo mitotic slippage. This is supported by the G_2_/M accumulation of MDA-MB-231 TNBC cells after treatment with zampanolide, similar to the effects of paclitaxel (Figure 3C). Together, these data demonstrate that zampanolide promotes microtubule stabilization, leading to the mitotic arrest and antiproliferative efficacy in a molecularly diverse subset of TNBC cells in a manner similar to other MSAs.

### 2.3. Zampanolide Has Persistent Efficacy In Vitro

While zampanolide and paclitaxel have similar effects on biochemical tubulin polymerization (Figure 1) and antiproliferative potency in TNBC cells when the drug remains in the media throughout the 48 h assay period (Figure 2, Table 1), we anticipated that the irreversible binding of zampanolide to tubulin would provide superior in vitro efficacy when the drug was only available for a short period prior to being washed off. This would be relevant to the in vivo situation where the drug is actively cleared. Moreover, a strong degree of cellular persistence following the removal of excess drug from the medium has been associated with potent in vivo antitumor efficacy for other classes of microtubule binding drugs, including eribulin and the taccalonolides [15,20]. To evaluate cellular persistence in vitro, we treated MDA-MB-231 cells at a low density with 10 or 50 nM zampanolide or paclitaxel for 2 h after which the drug containing media is washed off and replaced allowing cells to grow into colonies without any exogenous drug for an additional 10 days. Just a 2 h incubation of MDA-MB-231 cells with as little as 10 nM zampanolide was sufficient to completely inhibit colony growth over the subsequent 10 day period (Figure 4). In comparison, 10 nM paclitaxel had no effect on the number of colonies formed as compared to vehicle treated cells and 50 nM had an intermediate, but non-significant effect on colony growth (Figure 4). The cellular persistence of zampanolide is reminiscent of the activity of the taccalonolides in this assay and demonstrates the potential advantage of zampanolide over paclitaxel as a covalent stabilizer not only to circumvent drug resistance mechanisms, but also to provide long-term efficacy after acute treatment in both drug sensitive and resistant TNBC.

### 2.4. Antitumor Efficacy of Zampanolide

In spite of numerous reports of the in vitro efficacy of zampanolide in both drug sensitive and resistant cancer cell lines, there are no reports of in vivo antitumor evaluations of this compound, likely due to limited supplies of this complex natural product by either biosourced [5,6] or synthetic routes. However, due to the high degree of in vitro cellular persistence observed with zampanolide (Figure 4), we anticipated that it may be more potent in vivo than paclitaxel, allowing us to conduct an antitumor trial with small amounts of material. Indeed, we found that the maximum tolerated dose (MTD) of zampanolide in vivo in female athymic nude mice was less than 1.0 mg/kg total dose by intraperitoneal (i.p.) injection. We first evaluated the antitumor efficacy of zampanolide in a MDA-MB-231 human xenograft model in female athymic nude mice. A single systemic i.p. dose of 1.0 mg/kg zampanolide led to toxicity with up to 10% weight loss, and death in 50% of the animals 7 days post-injection. This indicates that zampanolide is bioavailable and over 100-fold more potent than docetaxel with regard to MTD [21]. However, zampanolide was determined to have no therapeutic window for antitumor efficacy with systemic administration as no decrease in tumor volume was observed at this LD_50_. Thus, another tumor trial was conducted with localized intratumoral injections of zampanolide to determine whether it could promote antitumor efficacy if it was able to reach the tumor without systemic toxicity. Animals bearing MDA-MB-231 xenografts were given direct intratumoral injections of 15 µg zampanolide or 3 µL DMSO vehicle in 100 µL PBS on days 0 and 7 (Figure 5A). A total dose of just 30 µg zampanolide caused a significant decrease in tumor volume as compared to vehicle controls. A divergence in tumor growth between the two groups was seen at day 5 with a significant difference in tumor growth by day 12 and increasing significance at days 14 and 16 (Figure 5B,C). Zampanolide-treated tumors remained under 1000 mm^3^ while vehicle treated tumors reached endpoint tumor volumes, leading to the end of the tumor trial on day 16 (Figure 5D). Even 9 days after the final dose, zampanolide tumors were notably smaller with a significant difference compared to vehicle-treated tumors (Figure 5E). These results demonstrate that zampanolide has potent and persistent antitumor efficacy with low doses maintaining antitumor efficacy over an extended period in this rapidly growing tumor model.

## 3. Discussion

Zampanolide is a microtubule stabilizing agent that has a unique mechanism of action from either taxanes or the taccalonolides that involves covalent binding within the taxane site on β-tubulin and stabilization of the M-loop to promote rapid and irreversible microtubule stabilization. We confirmed the mechanism of action of our purified zampanolide as a potent microtubule stabilizer through biochemical tubulin polymerization assays as well as immunofluorescence imaging and cell cycle analysis. Unsurprisingly, the potency and efficacy of zampanolide is comparable to paclitaxel in a panel of molecularly distinct TNBC cell lines with continuous 48 h drug exposure. However, zampanolide was markedly distinct from paclitaxel in its degree of cellular persistence such that a short, 2 h, treatment with 10 nM zampanolide was sufficient to completely inhibit the colony forming capability of TNBC cells over the subsequent 10-day period even after excess drug was removed from the medium. We hypothesized that this high degree of cellular persistence would translate to potent and persistent in vivo antitumor efficacy, which prompted the first ever in vivo evaluation of this irreversible microtubule stabilizer. Indeed, zampanolide is bioavailable and highly potent in vivo with an MTD of less than 1 mg/kg. However, it has an unacceptable therapeutic window for antitumor efficacy with systemic dosing, prompting evaluations of antitumor efficacy with intratumoral injection. As little as 30 µg of zampanolide provided antitumor efficacy for at least a week after the final dose. This finding demonstrates that zampanolide has potent and persistent antitumor efficacy when targeted to the tumor site. However, further work is needed to improve the therapeutic index of zampanolide, either by improving pharmacokinetic properties through medicinal chemistry efforts or by identifying mechanisms to improve its targeted delivery to the tumor.

There are multiple strategies that have been employed to improve the therapeutic window for microtubule-targeting agents (MTAs) in vivo. The most striking example has been the repurposing of highly toxic MTAs including the maytansine-derivative DM1 and the dolastatin-derivative MMAE as warheads for antibody-drug conjugates [22]. Additionally, the therapeutic index of paclitaxel can be further improved through the linkage of a 1,18-octadecanedioic acid, which forms a noncovalent complex with human serum albumin [23]. To take advantage of either of these approaches to improve the therapeutic index of zampanolide, it is critical to first understand its functional pharmacophore and potential chemical liabilities. While the chemically fragile side chain of zampanolide may be a liability, absence of this moiety in the metabolic precursor dactylolide and synthetic mimetics results in compounds with a significant reduction in cellular potency [10,24,25]. In contrast, single digit nanomolar potency can be retained upon replacement of the THP ring of zampanolide with a morpholine, which can serve as a convenient site for modifications to improve physiochemical properties and enhance tumor targeting while retaining binding within the taxane site [26]. Structure-activity relationship studies from zampanolide analogs isolated alongside the natural product demonstrate an unanticipated degree of flexibility in the macrolide core for retaining the potency and mechanism of action of zampanolide [27]. Over 45 analogs of this chemotype have been reported to date (mostly synthetic) [10,18,24,25,26,27,28] demonstrating that there is potential to modify zampanolide to improve its physiochemical and tumor-targeting properties.

Herein, we demonstrate that the exquisite in vivo potency of zampanolide offers the opportunity to explore the antitumor efficacy of zampanolide analogs with improved pharmacokinetic and tumor targeting properties even with small quantities of starting material. However, any derivatives that demonstrate an improved therapeutic window would still require a substantial increase in scale for true clinical development. The structure of this chemotype with only four chiral centers makes zampanolide amenable to total synthesis with results reported from multiple different research groups. To date, there have been over a dozen total synthesis of zampanolide and or variations of its macrolactone core ranging from 28 to 13 steps [29,30,31,32,33,34]. While these synthetic strategies are not trivial, the identification of a highly potent zampanolide derivative with an acceptable therapeutic window in vivo could provide the impetus to make the considerable synthetic efforts that led to the clinical success of the sponge-derived halichondrin derivative eribulin, which possesses 19 chiral centers and required 62 steps at the time of FDA approval [35]. In summary, zampanolide has a unique mechanism of action from any clinically approved MSA that allows it to circumvent drug resistance in vitro and provide potent and persistent in vivo antitumor efficacy, albeit with a narrow therapeutic window that will require further optimization through medicinal chemistry efforts to develop this chemotype for the treatment taxane-resistant disease.

## 4. Materials and Methods

### 4.1. General Experimental

HPLC purification of (−)-zampanolide was performed on a Luna 5 μm, C18(2) 100 Å 10 × 250 mm column (Phenomenex, Inc., Torrance, CA, USA) in conjunction with a 4.0 × 3.0 mm C18 (Octadecyl) guard column and cartridge (holder part number: KJ0-4282, cartridge part number: AJ0-4287, Phenomenex, Inc.). Compound detection was measured by means of a single wavelength (λ max = 230 nM) using an Applied Biosystems (Waltham, MA, USA) 759a UV detector. NMR experiments were performed on a Varian spectrometer (equipped with a 5 mm triple-resonance probe) at 400 MHz for ^1^H and 100 MHz for ^13^C experiments.

### 4.2. Isolation of (−)-Zampanolide

Samples of *C. mycofijiensis* were processed according to previous reports [36]. During the process of scale up isolation for fijianolide B (laulimalide), a repository fat-soluble dichloromethane-methanol extract (coded DMM), not previously processed due to its high lipophilic content, was further partitioned using 100 mL H_2_O (coded DMMW) and 100 mL dichloromethane (coded DMMF). The DMMF was evaporated and further partitioned with 100 mL 90:10 MeOH:H_2_O (coded DMMFM) and 100 mL hexanes (coded DMMFH) yielding a total of 315 mg DMMFM and 1169 mg DMMFH. Semi-preparative HPLC of the DMMFM extract (315 mg) using a reversed-phased gradient (30:70 CH_3_CN:H_2_O up to 80:20 at 50 min) yielded approximately 1.8% (−)-zampanolide from this extract, which was dried under a nitrogen stream, purged under argon, and stored in an amber vial in a dark desiccator under vacuum. NMR of purified (−)-zampanolide is available in the Appendix A. This compound is referred to as zampanolide throughout the article.

### 4.3. Cell Lines

Human TNBC cell lines MDA-MB-231, MDA-MB-453, HCC1806, HCC70, and HCC1937 were received from ATCC (Manassas, VA, USA). BT-549 cells were obtained from the Georgetown University Lombardi Comprehensive Cancer Center, Washington, DC. BT-549, HCC1806, HCC70, and HCC1937 were cultured in RPMI 1640 medium (Corning, Corning, NY, USA) supplemented with up to 10% FBS (Corning) and 50 μg/mL gentamicin (ThermoFisher Scientific, Waltham, MA, USA). MDA-MB-231 and MDA-MB-453 were cultured in modified IMEM (1X) supplemented with L-glutamine and up to 10% FBS and 50 μg/mL gentamicin. Cell line identity was authenticated by STR-based profiling (Genetica DNA Laboratories, Cincinnati, OH, USA). All cells were grown at 37 °C and 5% CO_2_ in an incubator and routinely tested for mycoplasma contamination.

### 4.4. Sulforhodamine B Assay

Cells were seeded into a 96-well plate (Corning) at an optimal density to maintain final readouts in the linear range of the assay. After adhering overnight, the cells were treated zampanolide at the indicated concentrations in 0.5% DMSO (Fisher Scientific, Waltham, MA, USA) for 48 h. A separate time zero plate was fixed with 10% trichloroacetic acid (Sigma-Aldrich, St. Louis, MO, USA) to provide a readout of cellular density at the time of drug addition. After treatment, the cells were fixed with 10% trichloroacetic acid for a minimum of 1 h. The cells were then stained with sulforhodamine B dye (Sigma-Aldrich) for 30 min. Excess dye was washed off and the cellular-bound dye was solubilized in 200 µL of 10 mM Tris before reading the optical density at 560 nm on the Spectramax plate reader running SoftMax Pro 5.4 (Molecular Devices, San Jose, CA, USA). The percent growth or cytotoxicity of cells during the treatment period was determined as compared to the time zero plate (y = 0) and vehicle treated wells (y = 100) and replicates graphed with error bars representing SEM. Concentrations that caused a 50% inhibition of cell growth (GI_50_) and total growth inhibition (TGI) were determined by non-linear regression analysis of the data using Prism (Graphpad, San Diego, CA, USA). Data are from 3 independent experiments each run in triplicate ± SEM.

### 4.5. Tubulin Polymerization Assay

The polymerization of purified porcine tubulin (Cytoskeleton, Denver, CO) was performed in GPEM buffer (80 mM PIPES pH 6.9, 2 mM MgCl_2_, 0.5 mM EGTA, and 1 mM GTP) in 10% glycerol. All components were kept on ice during preparation. Zampanolide, paclitaxel (Sigma-Aldrich), taccalonolide AF [13], or combretastatin A-4 (Sigma-Aldrich) were brought up as 2 mM stocks in DMSO. 1 µL of each of the 2 mM compound stocks were added to their respective individual wells of a 96-well plate in GPEM buffer to give a final concentration of 20 µM in 100 µL. 20 µM tubulin in GPEM was added immediately to each well before inserting the plate to be read by a pre-warmed 37 °C SpectraMax microplate reader (Molecular Devices, San Jose, CA, USA). Microtubule polymerization was monitored by measuring the change in absorbance at 340 nm every minute for an hour. Data were recorded and analyzed using SoftMax Pro 5.4 (Molecular Devices, San Jose, CA, USA). Data are from two independent experiments with error bars showing SEM at each time point.

### 4.6. Cell Cycle Analysis

MDA-MB-231 cells were seeded in 6 well plates at a density of 1 × 10^6^ cells/mL. After adhering overnight, the cells were treated with 10 nM or 50 nM paclitaxel, zampanolide or DMSO vehicle control for 20 h at indicated concentrations. Cells were harvested by scraping, washed with PBS, centrifuged at 450 g for 5 min and resuspended in 300 µL Krishan’s reagent (0.05 mg/mL propidium iodide, 1 mg/mL sodium citrate, 0.02 mg/mL ribonuclease A, 0.3% IGEPAL). Cell cycle distribution was analyzed based on cellular propidium iodide incorporation by flow cytometry using a FACS Calibur (BD Biosciences, Franklin Lakes, NJ, USA) and FlowJo software (FlowJo, Ashland, OR, USA).

### 4.7. Persistence Assay

MDA-MB-231 cells were diluted to a concentration of 100 cells/mL to seed 200 cells per plate with 2 mL of media in 60 cm dishes. After adhering overnight, cells were treated with 10 nM or 50 nM zampanolide or paclitaxel for two hours. The plates were gently rinsed with PBS before replacing with 2 mL of fresh IMEM+ 5% FBS media. The cells were left to grow into colonies for 10 days before fixing with 0.5% crystal violet in 20% methanol and visible colonies counted from two independent replicates. Significance between was determined using a one-way ANOVA with Sidak’s posthoc test for multiple comparisons.

### 4.8. Indirect Immunofluorescence

HCC1937 cells were seeded on coverslips in 24-well plates (Corning) and left to adhere overnight. Cells were treated with either 100 nM zampanolide or DMSO vehicle control for 18 h and then fixed with 100% methanol for 5 min. Indirect immunofluorescence was used to detect β-tubulin using a primary (Sigma-Aldrich, T-4026, 1:400) and a secondary conjugated to FITC (Sigma-Aldrich, F-3008, 1:200) and stained with 0.1 μg/mL DAPI (Sigma-Aldrich). Images were acquired on an i80 fluorescence microscope (Nikon, Tokyo, Japan) by compressing multiple y-stacked images using NIS elements software (Nikon).

### 4.9. Animals & Tumor Trial

Six-week-old female athymic nude mice were purchased from Envigo (Indianapolis, IN, USA). Tumors were established by injecting MDA-MB-231 tumor fragments bilaterally into the flank until the mean tumor volume reached 200 mm^3^ and all tumors included in the trial were greater than 50 mm^3^. For systemic dosing, a dose of 1.0 mg/kg zampanolide was given by i.p. injection in vehicle (*n* = 7 mice, 13 tumors) and compared to vehicle treated controls (*n* = 5 mice, 10 tumors). For local intratumoral injections, mice were given two doses of 15 μg zampanolide in vehicle (*n* = 5 mice, 10 tumors) on days 0 and 7 and compared to animals treated intratumorally with vehicle alone (*n* = 6 mice, 11 tumors). Tumor volume was measured by calipers (length × width × height) three times a week and weight monitored throughout the trials until day 16 when the first tumor reached the endpoint volume of 2000 mm^3^ in accordance with our IACUC protocol. Statistical differences between the vehicle and zampanolide treated groups were determined by a two- way ANOVA (drug * time) with Sidak’s posthoc test for multiple comparisons. All tumors were collected and weighed at this endpoint and significance between the vehicle and zampanolide treated groups determined by an unpaired *t*-test.

## Figures and Tables

**Figure 1 molecules-27-04244-f001:**
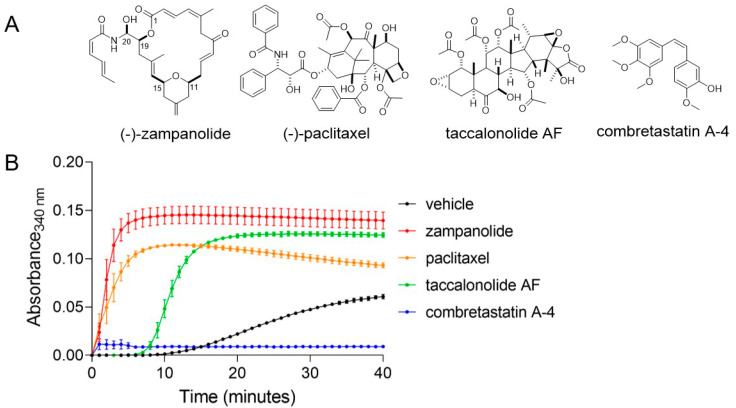
(**A**) Chemical structures of compounds. (**B**) Polymerization of purified porcine tubulin (20 µM) with DMSO vehicle or 20 µM zampanolide, paclitaxel, taccalonolide AF or combretastatin A-4. Polymerization was measured by the absorbance at 340 nm every minute for 40 min. *N* = 2 ± SEM.

**Figure 2 molecules-27-04244-f002:**
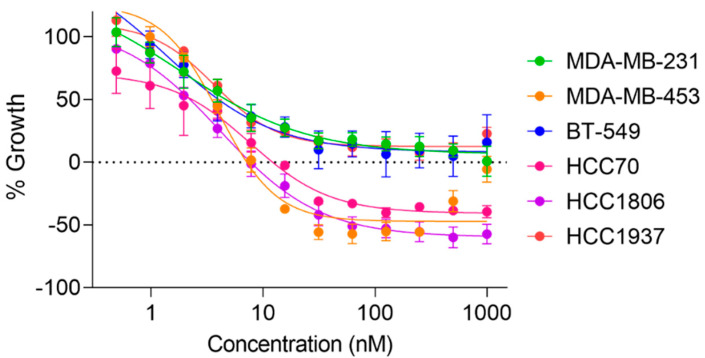
Concentration-dependent antiproliferative and cytotoxic effects of zampanolide in a panel of TNBC cell lines. The cell density at the time of drug addition is represented by the dashed line at y = 0 such that cytotoxicity is represented by values that drop below this line. Analysis with non-linear regression log(agonist) vs. response–variable slope. *N* = 3 ± SEM.

**Figure 3 molecules-27-04244-f003:**
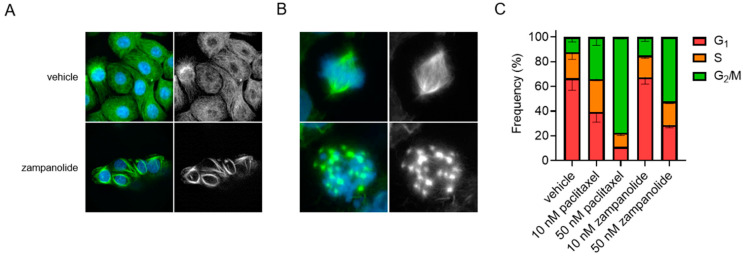
Microtubule stabilizing and antimitotic effects of zampanolide in TNBC cells. (**A**,**B**) Immunofluorescence images of microtubules in interphase (**A**) or mitotic (**B**) HCC1937 TNBC cells treated with zampanolide at 100 nM or vehicle control for 18 h. Microtubules are green and DAPI nuclear stain is blue. (**C**) MDA-MB-231 cells were treated for 20 h with 10 or 50 nM paclitaxel or zampanolide and cell cycle distribution evaluated by flow cytometry of cells stained with propidium iodide. *N* = 2 ± SEM.

**Figure 4 molecules-27-04244-f004:**
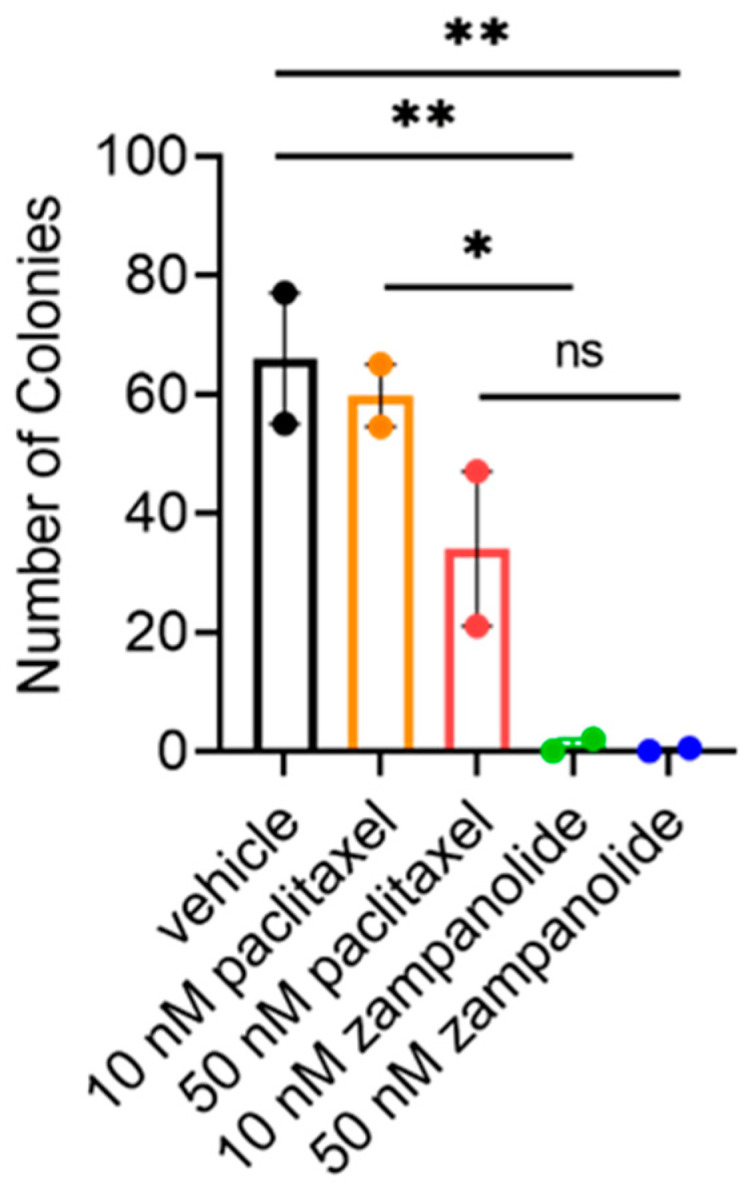
Zampanolide has superior cellular persistence compared to paclitaxel in MDA-MB-231 cells. Quantification of colonies formed by MDA-MB-231 cells 10 days after a 2 h treatment with 10 or 50 nM paclitaxel or zampanolide prior to drug washout. Analysis conducted with one-way ANOVA with Sidak’s multiple comparisons. *N* = 2 ± SEM. * *p* < 0.05, ** *p* < 0.01, ns = no significance.

**Figure 5 molecules-27-04244-f005:**
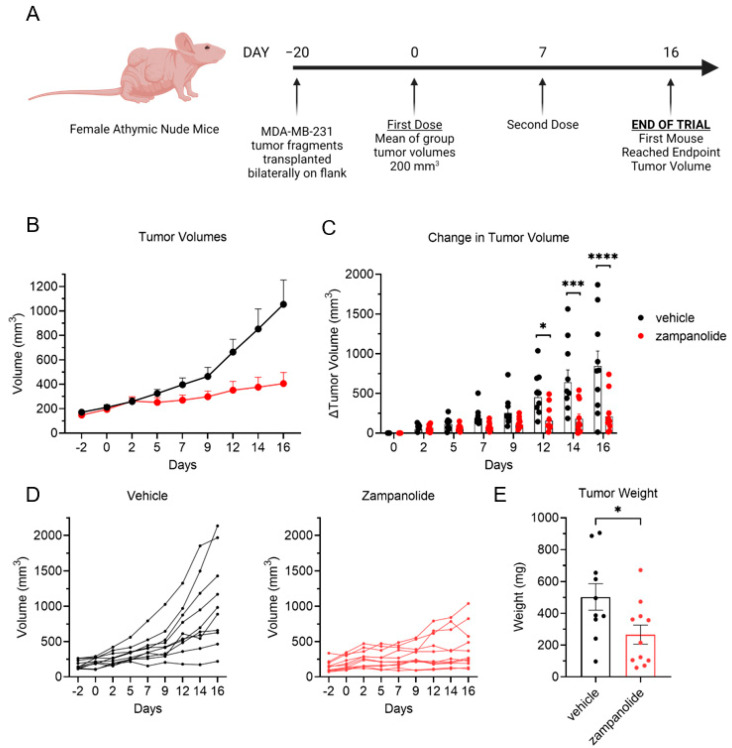
Antitumor efficacy of zampanolide in MDA-MB-231 TNBC tumors. (**A**) Timeline of tumor trial. Female athymic nude mice with bilateral MDA-MB-231 tumors were injected intratumorally with 15 µg zampanolide (*n* = 10) or 3 µL DMSO in 100 µL PBS vehicle (*n* = 11). (**B**) Tumor volumes of vehicle and zampanolide groups over time as average ± SEM. (**C**) Change in tumor volume of vehicle and zampanolide groups over time as average ± SEM showing individual values. Significance determined by two-way ANOVA with Sidak’s multiple comparisons. (**D**) Spider plot of individual tumor volumes from vehicle and zampanolide treated animals. (**E**) Final weight of each tumor from vehicle and zampanolide treated animals. Significance determined by unpaired T-test. * *p* < 0.05, *** *p* < 0.001, **** *p* < 0.0001.

**Table 1 molecules-27-04244-t001:** GI_50_
^1^ values (nM) calculated from zampanolide dose–response curves for each TNBC cell line. *N* = 3 ± SEM.

	MDA-MB-231	MDA-MB-453	BT-549	HCC70	HCC1806	HCC1937
zampanolide	5.4 ± 1.5	3.8 ± 0.3	4.4 ± 0.7	4.1 ± 0.4	2.8 ± 0.5	5.3 ± 0.9
paclitaxel	3.3 ± 0.6 ^2^	-	2.8 ± 0.7 ^2^	-	0.9 ± 0.3 ^2^	-

^1^ GI_50_ is the concentration that causes 50% inhibition of growth over the 48 h treatment period. ^2^ Reprinted from [17] under creative commons license https://creativecommons.org/licenses/by-nc-nd/4.0/ at https://pubs.acs.org/doi/abs/10.1021/acsomega.1c07146 (accessed on 14 May 2022) with no changes.

## Data Availability

Data is contained within the article and Appendix A.

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
