# Peer review of "In Vivo Evaluation of (−)-Zampanolide Demonstrates Potent and Persistent Antitumor Efficacy When Targeted to the Tumor Site"

_molecules, 2022, doi:10.3390/molecules27134244_

Round 1
Reviewer 1 Report
Overall assessment
This manuscript describes additional biological evaluations of the natural MSA zampanolide, including the first in vivo experiments with tumored mice. The compound turns out to be potent, persistent and toxic, indicating that clinical development will face the double challenge of compound acquisition and a highly specific dosing approach. Overall the article presents important new results and is publishable after the corrections noted below are made.
Editorial comments
P2 l 85 “Herein we describe the isolation of an unanticipated source of zampanolide.” This statement may be technically correct, but the discovery is buried in the experimental section and no evidence is given that C. mycofijiensis is in fact a better source than that described in previously published work. A clearer presentation should include these additional details, specifically comparing yields from C. mycofijiensis and the original Fasciospongia rimasa source.
P9 l 313-315 After treatment, the cells were fixed with 10% trichloroacetic acid and protein stained with sulforhodamine B dye (Sigma-Aldrich) which was then solubilized in 200 µL Tris before reading the optical density at 560 nm on the Spectramax plate reader running SoftMax Pro 5.4 (Molecular Devices, San Jose, CA). Was sulforhodamine B dye (Sigma-Aldrich) solubilized in 200 µL Tris? That is how the sentence reads
P9 l 326-331 The mammoth sentence “Zampanolide, paclitaxel (Sigma-Aldrich), taccalonolide AF [13], or combretastatin A-4 (Sigma-Aldrich) were brought up as 2 mM stocks in DMSO and 1 µL was added to the reaction to give a final concentration of 20 µM in 100 µL were added to individual wells of a 96-well plate in GPEM buffer and then a tubulin at a final concentration of 20 µM in GPEM was added immediately before inserting the plate to be read by a SpectraMax microplate reader (Molecular Devices, San Jose, CA) which was pre-warmed to 37ËšC.” does not make sense. What were added to individual wells? Rephrase into at least two clear separate sentences.
P10 l 382 replace 1H NMR with 1H NMR and 13C NMR with 13C NMR
Author Response
We thank the reviewer for taking the time to diligently review our manuscript. We appreciate the reviewer’s positive comments and feedback on our findings. We have made revisions on each of the sections listed by the reviewer to improve clarity.
P2 l 85 “Herein we describe the isolation of an unanticipated source of zampanolide.” This statement may be technically correct, but the discovery is buried in the experimental section and no evidence is given that C. mycofijiensis is in fact a better source than that described in previously published work. A clearer presentation should include these additional details, specifically comparing yields from C. mycofijiensis and the original Fasciospongia rimasa source.
We agree that since we do not directly compare yields to previous isolations, we have changed our language to remove ‘unanticipated’ and instead just indicate the identity of this source of zampanolide.
P9 l 313-315 After treatment, the cells were fixed with 10% trichloroacetic acid and protein stained with sulforhodamine B dye (Sigma-Aldrich) which was then solubilized in 200 µL Tris before reading the optical density at 560 nm on the Spectramax plate reader running SoftMax Pro 5.4 (Molecular Devices, San Jose, CA). Was sulforhodamine B dye (Sigma-Aldrich) solubilized in 200 µL Tris? That is how the sentence reads.
We separated the individual steps into their own sentences to clarify that excess dye was washed off and the cellular-bound dye was solubilized by 10mM Tris.
P9 l 326-331 The mammoth sentence “Zampanolide, paclitaxel (Sigma-Aldrich), taccalonolide AF [13], or combretastatin A-4 (Sigma-Aldrich) were brought up as 2 mM stocks in DMSO and 1 µL was added to the reaction to give a final concentration of 20 µM in 100 µL were added to individual wells of a 96-well plate in GPEM buffer and then a tubulin at a final concentration of 20 µM in GPEM was added immediately before inserting the plate to be read by a SpectraMax microplate reader (Molecular Devices, San Jose, CA) which was pre-warmed to 37ËšC.” does not make sense. What were added to individual wells? Rephrase into at least two clear separate sentences.
This section of the method has been separated into multiple sentences and reworded to make each step of the method clear.
P10 l 382 replace 1H NMR with 1H NMR and 13C NMR with 13C NMR
This has been corrected.
Reviewer 2 Report
In this study, authors evaluated the antitumor efficacy of zampanolide, a new microtubule-stabilizing compound, in a triple-negative breast cancer (TNBC) xenograft model. A non-inferiority of zampanolide as compared to approved drugs was demonstrated in a series of TNBC cell lines; however, zampanolide exert a more persistent in vitro activity compared to the paclitaxel. A consistently persistent antitumor efficacy was finally demonstrated in the rapidly growing tumor model.
Experimental plan was well developed and conducted; data are well presented and sound convincing in proposing zampanolide as a new drug able to circumvent drug resistance.
Author Response
We thank the reviewer for taking the time to diligently review our manuscript. We appreciate their positive feedback and support of our findings.